# OpenSWAP, an Open Architecture, Low Cost Class of Autonomous Surface Vehicles for Geophysical Surveys in the Shallow Water Environment

**Giuseppe Stanghellini** [1] , **Fabrizio Del Bianco** [2] **and Luca Gasperini** [1,*]

1   ISMAR-CNR, Istituto di Scienze Marine, U.O. Geologia Marina, CNR, Via Gobetti 101, 40129 Bologna, Italy; giuseppe.stanghellini@ismar.cnr.it
2   Proambiente Scrl, Via Gobetti 101, 40129 Bologna, Italy; f.delbianco@consorzioproambiente.it
*   Correspondence: luca.gasperini@ismar.cnr.it

**Abstract:** OpenSWAP is a class of innovative open architecture, low cost autonomous vehicles for geological/geophysical studies of shallow water environments. Although they can host different types of sensors, these vehicles were specifically designed for geophysical surveys, i.e., for the acquisition of bathymetric and stratigraphic data through single- and multibeam echosounders, side-scan sonars, and seismic-reflection systems. The main characteristic of the OpenSWAP vehicles is their ability of following pre-defined routes with high accuracy under acceptable weather and sea conditions. This would open the door to 4D (repeated) surveys, which constitute a powerful tool to analyze morphological and stratigraphic changes of the sediment/water interface and of the shallow substratum eventually caused by sediment dynamics (erosion vs. deposition), slumps and gravitative failures, earthquakes (slip along seismogenic faults and secondary effects of shaking), tsunamis, etc. The low cost and the open hardware/software architectures of these systems, which can be modified by the end users, lead for planning and execution of cooperative and adaptive surveys with different instruments not yet implemented or tested. Together with a technical description of the vehicles, we provide different case studies where they were successfully employed, carried out in environments not, or very difficultly accessed through conventional systems.

**Keywords:** Autonomous Surface Vehicles (ASV); marine geophysics; shallow water environments; repeated 4D surveys; NAIADI Project (New Autonomous/automatIc systems for the study AnD monitoring of aquatic envIronments)

## 1. Introduction

Natural or artificial shallow water environments, such as harbors, coastal areas, waterways, lakes and lagoons, are in general affected by anthropogenic pressures. For this reason, they would require periodic monitoring, to mitigate the effects of environmental crises caused by human activity or natural processes. However, to date, geophysical studies in shallow water areas (shallower than a few meters) are not a consolidated practice for various reasons, including the following: they present difficult access, even using small boats, in absence of accurate bathymetric maps, the shallow water represents an efficient waveguide for acoustic and ultrasonic noises that limits penetration of the signals into the substrate and the quality of echographic and seismic data, the effect of noise due to propellers, or other natural and artificial causes, is amplified, and the rapidity of environmental changes would require repeated investigations (4D), which is not economically viable with conventional methods.

The economic and social importance of shallow water environments, therefore, calls for the development of new technologies and methods, that could open their study to a wider range of

researchers and environmental protection agencies: progresses and developments in the field of marine robotics could be an interesting opportunity to achieve this goal. In fact, the relatively recent availability of miniaturized although accurate sensors, as well as the development of innovative hardware architectures (Arduino®, Raspberry™, etc.) simplify design and implementation of low cost but highly performing Autonomous Surface Vehicles (ASV), which can operate in a variety of aquatic environments. This is the case of SWAP (Shallow Water Autonomous Prospector) a class of vehicles developed by ISMAR-CNR and Proambiente Scrl, characterized by limited size, high versatility, and low cost, and operating with a variety of different payloads. OpenSWAP is the follow-up of that original project, developed under NAIADI (New Autonomous/automatIc systems for the study AnD monitoring of aquatic envIronments, https://www.consorzioproambiente.it/en/projects/terminati/51-naiadi-por-fesr-2014-2020), in the frame of a POR-FESR Emilia Romagna initiative [1]. The intensive use of "open" technologies and software packages for data acquisition and processing [2–4], as well as the low cost of production, have the potential to extend the use of these techniques and methods to a growing public of scientists studying geological processes in these rapidly changing environments. Although to date several ASV are available on the market, we believe that OpenSwap is innovative in many respects, as summarized in Table 1.

**Table 1.** Main characteristics of the OpenSwap vehicles.

| |
|---|
| Open HW/SW architectures, widely documented and easily modified by the end-users |
| Innovative hulls design which optimize navigation and data acquisition |
| Embedded geophysical instrumentation, including a single-beam echosounder (with full echogram recording), and a "chirp" sub-bottom profiler |
| High accuracy (within +/−30 cm) in repeating programmed navigation lines |
| Modular HW/SW interfaces which enable for integrating thirty-party instruments, both for navigation and data acquisition |

Together with a technical description of the vehicles, we present here some examples of data acquisition, which include single-beam echograms, side-scan sonar images, seismic reflection profiles, as well as multibeam data from different shallow water environments.

## 2. OpenSWAP, Philosophy and Motivations

OpenSWAP is a class of ASVs (autonomous surface vehicles) developed with the aim of providing flexible and easily operating autonomous aquatic vehicles (Figure 1), from both hardware and software point of view, allowing to perform data acquisition in the shallow water environment.

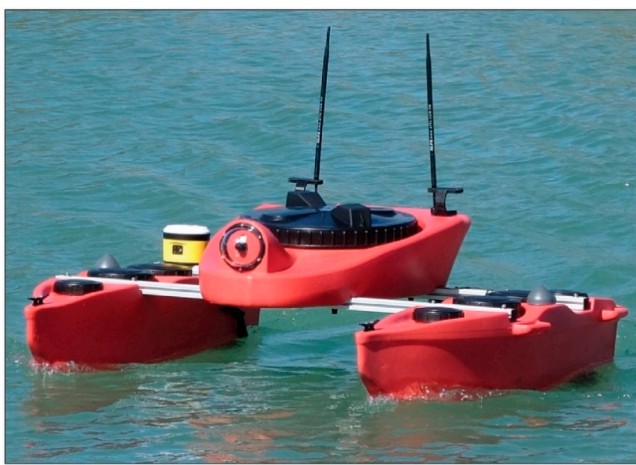

**Figure 1.** An OpenSWAP vehicle during a test phase.

Although these ASVs are suitable for a number of different payloads, including video cameras, current-meters, chemical and physical water sensors, water samplers, etc., we focused their design on acquisition of marine bathymetric and seismo-stratigraphic data. Our first target was implementing the ability of performing repeated surveys, i.e., following navigation paths with centimeters accuracy during subsequent runs. In fact, such performances are mandatory to analyze and monitor time-variant environmental processes and variables. Another functionality considered important was the possibility of planning in advance every practical aspect of a survey, such as: type of sensor employed and coverage to be obtained in a study area; routes to be followed by the vehicle during the run, avoiding obstacles; acquisition parameters (data sampling rate, maximum depths, etc.); time needed to complete the survey, in relation to batteries duration. This would allow for optimizing workflows in logistically difficult environments, reducing risks and deployment time. The development of the OpenSWAP vehicles was focused from the very beginning considering implementation of two embedded geophysical sensors: (1) a single-beam echosounder (SBE), to perform bathymetric (repeated) surveys, and (2) a chirped sub-bottom profiler (SBP), allowing for the acquisition of high-resolution stratigraphic data.

The vehicle was designed small enough to be easily transported and deployed, also in difficult conditions (steep shores; absence of docks; waters too shallow; etc.), but large enough to host suitable payloads and batteries. In a short summary, the design of the ASV was performed considering the following characteristics: housing as many batteries as possible, in relation to floating and navigation performances, to ensure long, self-running surveys with minimum returns to the base point; hosting in a safe waterproof container the on-board electronics; gathering implementation of different propulsion systems (i.e., waterborne vs. aerial propellers); providing additional space for hosting most common third-party sensors. Other features considered important were: low acoustic noise and low water turbulence induced by propellers and hulls close to the acoustic sensors; compatibility with AI (artificial intelligence) stay-on-route algorithms, to gather repeated surveys with centimeters errors, also in presence of water and air turbulence caused by waves, currents, and/or strong wind under acceptable weather and sea conditions. During our tests, we have found that navigation paths are followed up to Beaufort 4, but data quality under such conditions is very poor. We conclude that Beaufort 3 would be the limit for single-beam acoustic surveys.

The final version of the vehicles should have been easily customizable, allowing for installation of additional proprietary sensors, such as commercially available multibeam echosounders (MBES), SBP, water samplers etc.

To obtain such performances, the main electronic board has been developed to include several low-level serial RS232 I/O ports, as well as higher level network supports (i.e., Ethernet, Wi-Fi etc.), allowing for interfacing any proprietary instruments with the in-house positioning systems in the frame of a local-area network architecture.

The software and firmware routines were developed in form of separate modules, to allow the end-users implementing new functionalities through the use of internal scripting. This feature was improved to enable low-level software customizations, including new navigation algorithms, to enhance, for example, navigation accuracy under special condition/environment, or to adapt the current course to incoming parameters (wind speed and direction, wave height, etc.). Finally, the choice of an open hardware architecture for the electronics allowed to reduce costs, favoring distribution of those parts not available on the consumer market, also in the form of self-assembled kits (see next section).

## 3. Design of the Vehicles

### 3.1. Nautical Aspects

For several reasons, we decided to base the design of the vehicles on a multi hull scheme, and in particular a catamaran. In fact, compared to mono hull boats, catamarans are characterized by: wider beams, which increase stability; shallower draught, a major plus in shallow waters; smaller hydrodynamic resistance, implying less power delivered to the propellers. The design was carried

out also considering some specific requirements of geophysical surveys, which employ acoustic and ultrasonic signals generated and received by transducers that should be perfectly coupled with the water. For these reasons, the vehicles would have been able to minimize the tradeoff between "speed of the vehicle", which influences the time required to complete a survey, and "low acoustic noise in the water", caused by turbulence generated by propellers and viscous drags along the hull. The first technical solution to limit this negative effect, particularly evident in shallow waters, was installing the acoustic sensors between the two hulls of the catamaran, in an area characterized by relatively low water turbulence and shielded by the bubble carpet produced by propellers.

The main task of the OpenSWAP vehicles is performing geophysical surveys for study morphology and stratigraphy of the seafloor and penetrate the first tens of meters in unconsolidated or poorly consolidated sediments. For such objectives, acoustic and ultrasonic transducers are generally used, with alternating emission and detection of acoustic signal towards the bottom and below. Two instruments constitute very basic sensors for such surveys: the SBE ($10^1$–$10^2$ kHz), for accurate determination of depth and bottom reflectivity; the SBP, generating lower frequency ($10^0$–$10^1$ kHz) impulsive or frequency-modulated (chirped) signals, penetrating the subbottom and being reflected by acoustic impedance contrasts.

These instruments, particularly the SBP, are very sensitive to acoustic and electric noise, as well as to the presence of air bubbles in the water originated by turbulence close to the hulls or by propellers. To minimize such effects, we developed a catamaran with asymmetric hulls. Once designed, the fluid dynamic behavior was CFD (computational fluid dynamics) simulated, to iteratively optimize their shapes and to determine positions affected by minimum noise. As shown in Figure 2, a CDF plot of turbulence simulated at 7 km/h (3.7 knots) speed, the region between the hulls is very silent, and hence the best place for deploying the acoustic transducers. At higher speeds, noise slightly increases, but the region between the hulls remains the most favorable.

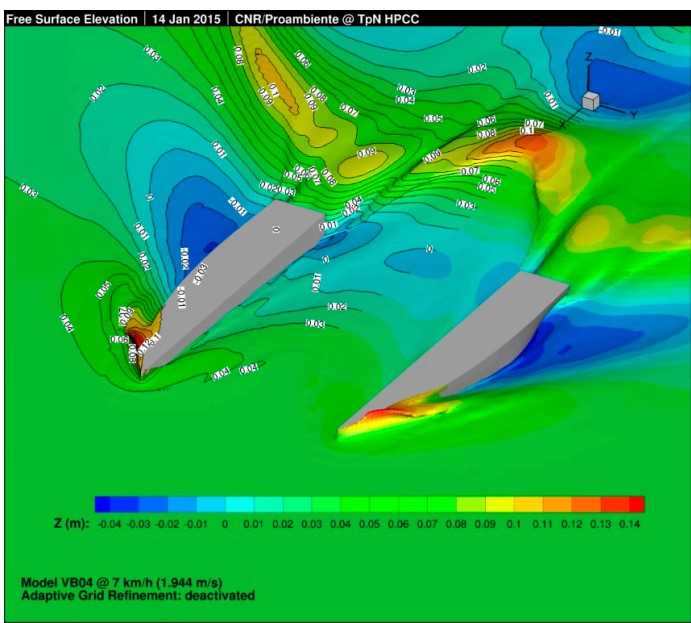

**Figure 2.** CFD simulation of the hulls behavior during the design phase of the vehicle at 7 km/h of speed. Note minimum turbulence between the two hulls.

The drawback of this design is that a catamaran with asymmetric hulls shows a slight stronger friction relative to the symmetric ones, but a very low noise between the hulls (Figure 2).

The design of hulls and vessel containing the electronics was carried out through the implementation of different standalone vehicles, whose tests carried out during different acquisition trials led to a final design implementing an electric powered plastic catamaran, made by linear

low-density polyethylene (LLDPE) with two asymmetrical lateral hulls and a central case housing the electronics (Figure 3). The frame is made of aluminum profiles, which provide connection between different parts of the vehicle and could be used as supports to deploy other instruments and sensors within the low noise area between the hulls.

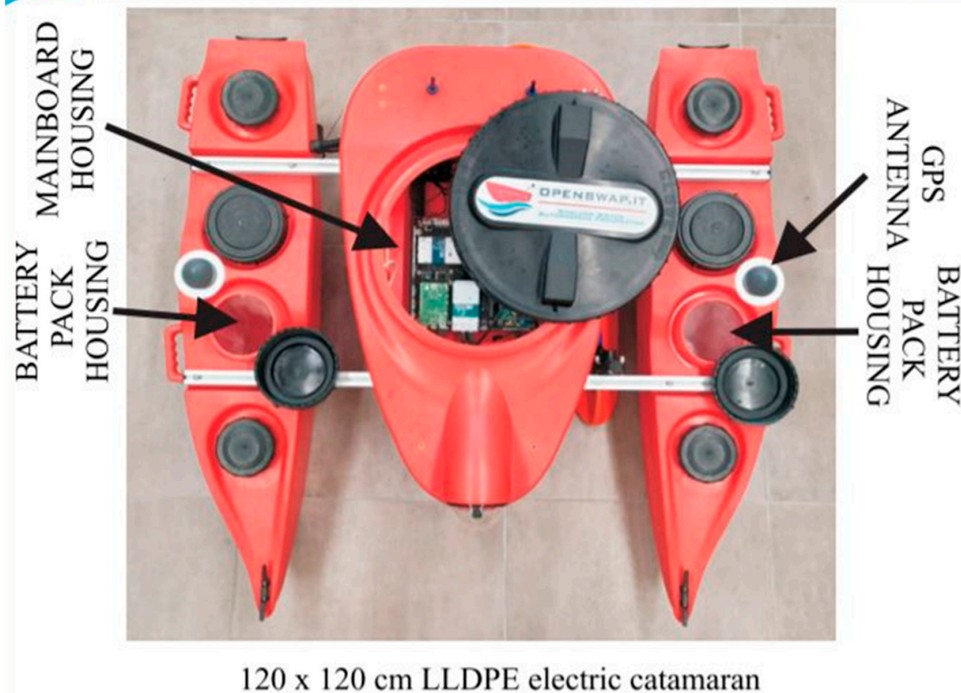

**Figure 3.** Photo and technical details of the latest OpenSWAP vehicle.

### 3.2. Electronic Boards

The "core" of the OpenSWAP hardware is constituted by a single-board multilayer PCB (printed circuit board), displayed in Figure 4, designed and developed to host embedded subsystems and boards (Arduino®, Raspberry™, Xbee, etc). It provides all connections and power supplies to electronic boards and devices, as well as signal conditioning circuits and I/O ports. Several jumper pins are included to quickly modify settings and parameters according to special setups and/or payloads. This is the only electronic component not available in the consumer market and distributed by Proambiente (https://www.openswap.it/).

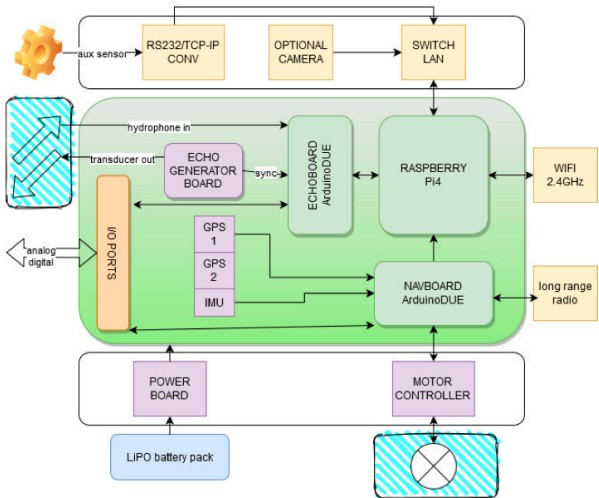

**Figure 4.** *Cont.*

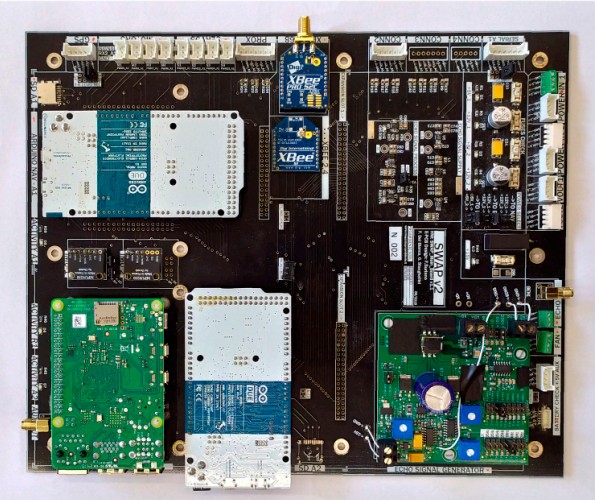

**Figure 4.** Block-diagram (**top**) and photo (**bottom**) of the OpenSWAP mainboard.

### 3.3. Software Architecture

The software routines which control OpenSWAP are divided into two main groups, those for navigation and those for data acquisition. A block-model of the software architecture is displayed in Figure 5, and described in detail below.

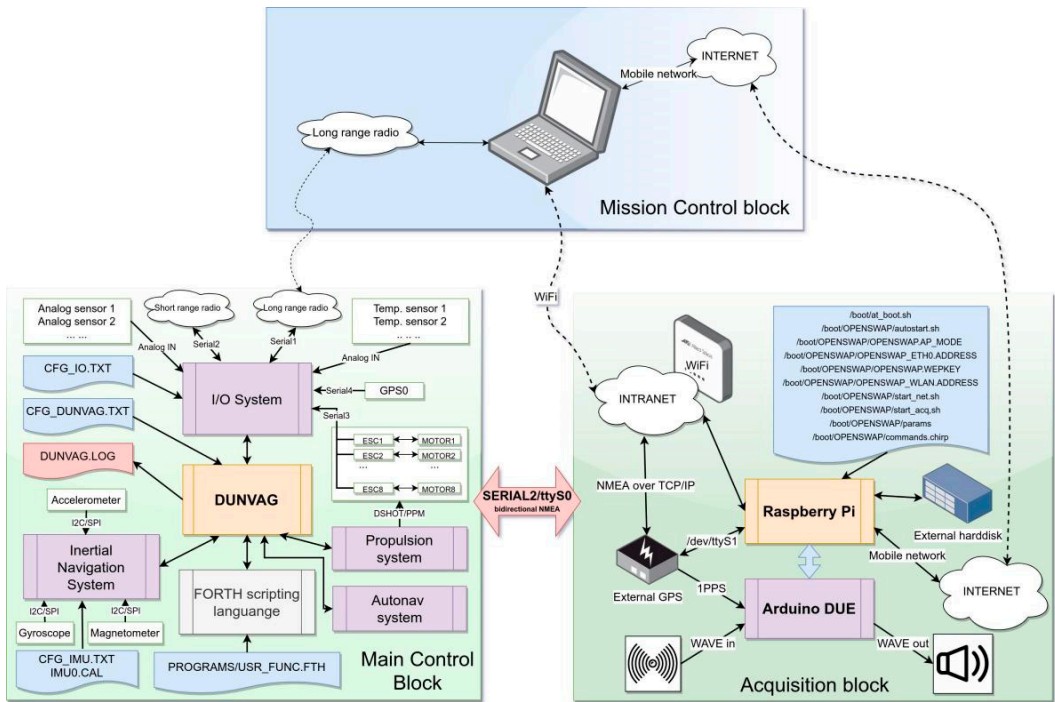

**Figure 5.** Block diagram showing OpenSWAP hardware/software architecture.

### 3.3.1. DUNVAG, the Autonomous Navigation Firmware

DUNVAG represents the "heart" of the system. It is a multiplatform firmware that runs on several different commercially available micro-controllers (i.e., ArduinoDue®, Teensy, STM32, etc.), and implements several functions, including: full control over propulsion system by using ESC (electronic speed control) boards; inertial navigation (INS), by means of MEMS (micro electro-mechanical systems) gyroscopes, accelerometers, magnetometers and single or double GPS antenna (with or without RTK correction); several algorithms for autonomous navigation and path following; wireless

communication for telemetry and remote control, both for long range point-to-point connections, and by means of wide area links using 3G/4G/5G terminals. DUNVAG is configured through the editing of three text files stored inside a μSD card at the root level: CFG_NAV.TXT for configuring navigation, CFG_IMU.TXT, for the inertial system, and CFG_IO.TXT for I/O ports. A detailed description of all parameters is reported in Supplementary Tables S1–S3.

The module controlling propulsion and direction changes of the vehicle is implemented through PWM (pulse width modulation) or DSHOT1200/2400 protocols, driving two (or more) underwater electrical motors connected to STM32 based ESCs. The system is able to process feedbacks from the ESCs (when DSHOT protocol is used) to inspect the status of active motors, such as: -current absorption; -applied voltage; spinning velocity (RPM); and ESC temperature. The module is also responsible for controlling direction changes during navigation by modulating the power distributed between port and starboard. Typically, the OpenSWAP vehicles mount two motors per side, a redundancy which results critical in case of failure, but also useful for enhancing propulsion under difficult environmental conditions (strong currents, waves, wind, etc.). Since the propulsion module supports up to eight motors, and because four propellers are generally used, the spare slots are available for customizations, including implementation of different types of propulsion. An example, that was specifically tested for wetlands and swamps, is implementing an airboat by installing air propellers on top of the vehicle to avoid aquatic plants and reduce the environmental impact. One specific feature implemented in this module, is the real-time acquisition of the propulsion devices telemetry, allowing for intervention in the event of partial or total failure. Through the analysis of the ESCs status, the propulsion module can monitor at a rate of $10^2$ samples per second propellers and controllers, and use such data to manage emergencies. A typical example is case of a propeller failure overcome by distributing more power to the other motors in the attempt of continue the navigation under the predefined constraints. This allows the vehicle to attempt accomplishing the mission in case of minor failures and/or re-entering the base in case o major. The configuration file including all parameters controlling the propulsion module is CFG_NAV.TXT (Supplementary Table S1).

The *I/O system* allows internal and external devices to be connected to the vehicle. It is based on several input ports, including eight RS232 serial lines and eight 0–5 V analogue inputs. In the default configuration, serial lines are configured to be connected to basic devices used for normal operations. They include:

(1) Serial-1, connected to the 433 MHz high-power, long-range, wireless module used for telemetry and remote controls;

(2) Serial-2, connected to the SBE module, is used to pass to the echosounder computer (a *Raspberry*™ PI) all the NMEA (National Marine Electronics Association) encoded sentences coming from the installed GPS, plus additional information in NMEA custom formatted strings, such as all vehicle status parameters, water temperature if temperature sensor is installed and any other readings coming from additional analog sensor eventually installed; asynchronously, *Raspberry*™ returns to the DUNVAG software the water depth computed in real time by analyzing the received echograms, all the NMEA sentences of any serial sensor connected at its ports and eventually the control-packets coming from the WAN (wide area network) detailed later on;

(3) Serial-3, connected in shared mode with every ESCs for telemetry;

(4) Serial-4, connected to the internal double RTK GPS receiver for high precision headings;

The remaining two serial lines are available for additional devices and can be accessed by the users through the internal FORTH scripting language.

Eight lines are also available for external analogue devices, accepting any input in the 0–5 V range, one already assigned to a temperature sensor embedded in the echosounder. Configuration parameters can be set to apply bias and/or scaling corrections if needed. Analogue lines could be also accessed by means of FORTH language commands, as well as through custom $IIXDR sentences sent to Serial-2. The sequence of such sentences is stored in a log file by the *Raspberry*™ echosounder module and/or eventually sent over the INTERNET when a connection is available.

The Inertial Navigation System is composed by three MEMS devices and one or two GPS receivers (see scheme in Figure 5). It is a key part of the system as it implements the HW/SW layer responsible for updating in real-time the vehicle position and its heading, also providing attitude information (pitch, roll and heave). It makes use of a MARG sensor fusion and Kalman 1D filtering algorithm [5–9] receiving data from gyroscopes, accelerometers and magnetometers, and integrating them with GPS heading and position. The inertial navigation system is configured through a text file, the CFG_IMU.TXT (Supplementary Materials), containing parameters for accelerometer and gyroscope bias, linear correction on temperature changes, and for hard and soft iron perturbation induced on magnetometer by external sources such as ferromagnetic material. Additionally, for more accurate calibrations over temperature, an IMU0.CAL file can be added, containing a list of corrections to be applied to gyroscope and accelerometer for a given temperature range ($-50/100$ °C). While a standard calibration is provided, in the event of calibration loss or unsatisfactory behaviors, an additional JAVA® application named MUVIMANT is available, to perform calibration of magnetometer and accelerometer/gyroscope couple.

Once calibrated, the system is able to integrate and correct the heading with using data from the GPS receiver, injecting the heading available through the IMU (Inertial Measurement System), prioritizing on most precise NMEA sentences (i.e., HDG before VTG). This provides a quick updating of the heading, since MEMS are characterized by a very fast and relatively accurate heading/positioning, and the GPSs, although slower, would provide best the accuracy in the presence of a good satellites signal.

Autonomous navigation system. Once accurate positioning and direction, as well as waypoints of the predetermined path are available to the system, they are sent to the autonomous navigation module (AIVAG), responsible for computing actual direction and power to be distributed to the propulsion module, which translate such indications into a direct control of the propellers by distributing the electrical power to the motors.

FORTH scripting language: Although it is well known that open standards and open applications are far better than the closed ones, there is still very few cases of real open applications giving to the users the possibility of "reprogramming the program", which would extend to unpredictable fields the application itself. This is mostly the case for marine oriented application, where the high software development costs do not leave additional economic resources to add extensions needed to open the application to the end-users. In most cases, not opening the software packages is among the politics of the software companies, operating in a small, very competitive market. However, in the field of Earth Sciences and more specifically in Marine Science there is a long-standing tradition for developing open software packages with high performances. Interesting examples of such packages presently diffused worldwide and used by a large audience of scientists are the Generic Mapping Tool [10] for maps and spatial data processing, MB-System [11], for processing MBES data, Seismic Unix [12] for seismic data processing, and many others. The OpenSWAP project would fall on the track of such scientific tools, and was developed assuming that all scientific applications developed by academic institutions should be, if not Open Source, at least open, giving the opportunity to adapt their functionalities to the scientific issue to be addressed. In this respect, DUNVAG implements a FORTH scripting language that gives the full control on almost every aspects of the application. It is based on a FORTH-83 implementation made available on the public domain by John Walker (https://www.fourmilab.ch/atlast/) and has been ported to DUNVAG to have access to most of the variables and parameters of the DUNVAG application. For manual of the language please refer to the above given URL. The built-in FORTH language is interfaced with DUNVAG through the execution of predefined FORTH commands, a block diagram of the DUNVAG program is shown in Figure 6, along with the FORTH entry points. Several variables of DUNVAG and some useful functions are available for full controlling the vehicle. Supplementary Tables S4 and S5 include a list of FORTH functions and variables.

**DUNVAG flow diagram**

**Figure 6.** Block model of *DUNVAG*, the hearth of the system which implement autonomous navigation algorithms.

Since execution of the FORTH routines is placed inside the main loop, it is very important that each call to those routines will not block the normal flow for too long. For this reason, they will be interrupted after 2 ms. of CPU time. Examples of FORTH programming are given into a download area, which also includes additional contents. The reader could find into those examples, amongst others, a definition that implements a virtual GPS through reading a file with lat, lon, pitch, roll, heading and speed values, allowing for a mission replay and/or simulation.

While the DUNVAG firmware is not yet open source (it will be in the near future), the FORTH scripting capabilities opens up the SWAP autonomous class of vehicles to user extensions, hence the name OpenSWAP.

### 3.3.2. The Embedded SBE/SBP Instruments

The block-diagram of Figure 5, shows how the SBE is implemented. It can be both integrated into the navigation board or available as a separate module, while the chirped SBP is available only as separate board. Both instruments are installed inside the water-proof central box, together with other electronics/sensors eventually added to the payload.

SBE and SBP follows the same design, based on a *Raspberry*™ PI driving an ArduinoDue® microcontroller. The ArduinoDue® is loaded with two small pieces of software, μEcho and μChirp, which use the microcontroller ADC (analog to digital) and DAC (digital to analog) ports, as well as some transmitting/receiving circuits/sensors, to emit and receive acoustic signals into the water in a given time-window. The Raspberry™ PI, on the others side, runs CIAPCIAP, the software responsible for geo-referencing the data coming from the ArduinoDue® and for its storing into the μSD installed card or into an external support (hard-disk). The SBE/SBP pair can work in parallel, leading for synchronous acquisition of stratigraphic and bathymetric data. It would be possible for example, to allow the two systems modify some acquisition parameters, such as pulse length, pulse frequency, acquisition time window, etc, based on analysis of incoming data, in an AI (artificial intelligence) scheme, relating their values to actual bottom depth or other different variable environment parameters.

μEcho/μChirp Both SBE and SBP systems, we name μEcho and μChirp, respectively, are based on an ArduinoDue® microcontroller, which emits analog signals through a DAC port sampled at a very high rate (<1 μs). This performance, together with the possibility of programming the microcontroller, allow for a complete control on the emitted signal, an important characteristic to optimize signal generation in relation to specific survey, and to optimize the deconvolution of the signal. All acquisition parameters are controlled by a set of commands sent to an USB port of the micro-controller, while a second port is used for data acquisition. The software module CIAPCIAP, running on a Raspberry™ PI,

is responsible for managing consistently these two ports, to implement a synchronized data acquisition. Supplementary Table S6 and S7 include the entire set of commands along with a brief description.

The μCʜɪʀᴘ SBP uses as emitter transducers lightweight electromagnetic resonators directly applied to the catamaran hulls, that show interesting performances in shallow water environments, and are easily installed on the vehicle. The system is composed by: (1) a digital generator of frequency-modulated signals based on an ArduinoDue® board; (2) a 600 W RMS power amplifier; (3) an array of waterproof magnetic resonators composed of four 4 Ω *MONACOR* transducers; (4) an acquisition system based on a hydrophone array, a signal amplifier, and an *ArduinoDue*® board used as analog-to-digital-converter (ADC). A Raspberry™ PI is employed to store the digital data in SEG-Y format [13] on a SD memory card.

The CIAPCIAP software module. CIAPCIAP runs on a Raspberry™ PI and is interfaced with DUNVAG (Figure 5) through a serial port to gather current position and orientation from the navigation board. On the other side, it passes to DUNVAG additional parameters such, as NMEA sentences from a GPS eventually connected to the geophysical sensors. CIAPCIAP makes also possible to connect to itself through a Wi-Fi hotspot client, or to implement an internal hotspot eventually used by external clients. It is also able to open a channel between the Mission Control Software (see next section) and DUNVAG by means of an INTERNET connection, eventually opened through available mobile networks. In such a case, a ssh tunnel with an external sshd server can be opened, allowing for connecting to the INTERNET from virtually everywhere. The operating system chosen for CIAPCIAP is Linux®, running from the installed μSD card in R/O (read only) mode for reliability reasons, and it is configurable by editing the files contained in the/boot/partitions (see Supplementary Table S8). The acquisition parameters are set by commands/parameters specified by the params, commands.echo and commands.chirp files, whose syntax is reported in the Supplementary Materials (Tables S6 and S7).

*CIAPCIAP* starts by reading the params file and subsequently opens the commands.echo or the commands.chirp files (the specified within params). It then sends all lines of the opened file to a proper USB port connected to the ArduinoDue® which set the emission and the acquisition parameters of the transmit/receive cycles. Such parameters are those controlling the cycle, such as the acquisition time window, the emitted waveform shape and length, the emission rate etc. Supplementary Tables S6 and S7, and could be modified by the user. This is obtained by editing a "commands file" containing initialization parameters for both, emitter and receiver. Special commands such as "WAVEOUT:" or "CHIRP:" (they are equivalent) are sent to the emitting or receiving processes. A configuration command sequence should begin with the receiver (sampler) configuration, followed by the emitter configuration. Once a WAVEOUT:MKSIGNAL command is sent, the emitting process adds a waveform with current parameters to a waveform buffer, along with current sampling parameters, allowing for the creation of waveforms of different length, duration and sweep. All waveforms are stored into an internal buffer and subsequently used to de-convolve the received signal. When the command WAVEOUT: GO is entered, the emission process starts, by generating the first available waveform since an emission timeout is reached. At this point, a second waveform (if present) could be generated, until the last waveform is reproduced. Then, the emission process starts again using the first available waveform. Synchronized with emission, the receiving procedure starts, digitally sampling the analog signal detected by the transducer/amplifier chain, according to parameters set for each configured waveform.

To start the whole emitting and receiving processes, the commands "WAVEOUT: GO" and "GO" should be entered in this order. After that, CIAPCIAP starts signal emission and the acquisition, and stops only if a "STOP" command and a "WAVEOUT: STOP" is sent.

An example of a commands file is given in Supplementary Tables S6 and S7, where a multi-chirp acquisition is activated with a start frequency of 24 Hz, and a 6 Hz frequency increasing every 4 generated waveforms. Each sweep generation is followed by data acquisition according to predefined acquisition parameters.

The acquisition process is not working separately by the other software modules. In fact, it is parallelized with several ancillary tasks dealing with: position and orientation updates (received from

an external GPS and/or from the internal NAVBOARD); managing connection to a remote monitoring window; receiving commands from remote controlling applications; checking periodically whether an INTERNET connection is available. In this case, CIAPCIAP connects to a proprietary server (openswap.it) opening a bidirectional channel that allows for a remote telemetry from DUNVAG and a basic control of the vehicle. Further details on such functionality are given in the next session.

### 3.4. OpenSWAPNav, the Mission Control Software

The Mission Control Software, OpenSWAPNav (OSN), written in Java®, was developed to allow for interactive route planning and vehicle remote control. It runs a small C++ [14] app, (SwapCONTROLLER) that communicates through a wireless long-range (433MHz) device with the vehicle, receives the telemetry at a rate of six times per seconds, and send to the vehicle a set user-command. It can receive inputs from the user by means of a joystick or through the user interface, easily allowing manual control of the vehicle. The choice to have a small app doing the basic jobs of communicating to and controlling the remote vehicle is due to the need of a quick start/restart control system in case of problems. The SwapCONTROLLER application allows to fully control the remote vehicle in manual or auto modes.

OSN is connected with SwapCONTROLLER through a socket, for vehicle telemetry and other information. This allows the user to easily edit the routes and send the waypoints to the vehicle. As seen previously, the vehicle needs that some configuration files are present into its μSD (CFG_IMU.TXT, CFG_IO.TXT, CFG_IMU.TXT etc.), and OSN provides download and upload such files to the vehicle to update all parameters. OSN can communicate with the vehicle also through the *INTERNET* (if the vehicle is configured for the purpose), can receive telemetry also through such a connection from the vehicle and send to commands, such as the next point to reach or where to stop navigation, or even maintain the actual position. This is particularly useful when the vehicle exits the coverage of internal 433 MHz telemetry module.

A Screenshot of OSN and SwapCONTROLLER during a survey is shown in Figure 7.

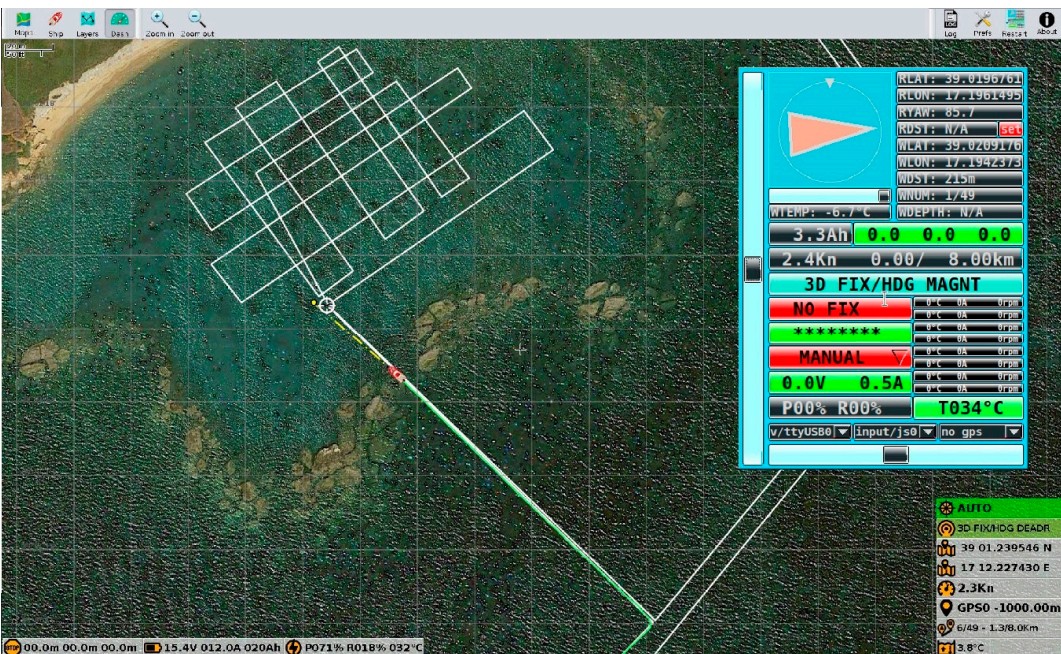

**Figure 7.** Screenshots of OSN and SwapController (**inset**) during a survey. Planned lines are marked in white, while navigation performed is indicated by green lines.

## 4. Performances of the Vehicles and Data Acquisition Examples

Performances and functionalities of the OpenSWAP prototypes were tested in several environments, under different weather and sea conditions, and different instrument payloads. Below, we report some examples of data acquisitions carried out in areas not, or very difficultly accessed by conventional surveys.

### 4.1. Navigation Accuracy

The accuracy in following pre-determined paths was first tested in "controlled" environments, such as small lakes, ponds and easily accessible coastal areas. Figure 8 shows an example of navigation test carried out in shallow waters along the Emilia Romagna coast, were two runs over the same planned lines were carried out. It includes a statistical evaluation of the navigation performances, i.e., of the errors that occurred in the following predetermined routes, which indicates that almost 90% of the route waypoints are within 0.30 m of polar distance from those planned. In worst cases, repeatability of the planned paths is within a mean error of about 0.40–0.50 m, reaching up to ~0.10 m in the best-case scenarios (calm weather and sea conditions). The test reported in Figure 8 consisted of navigating along a 1 km straight line towards the offshore during two different days. In this case, we used a double-antenna, double-frequency Trimble SPS461 GPS receiver, with an RTK cm accuracy. The mean error between the two path is 0.19 m, with a maximum over 99% of the points within 0.70 m.

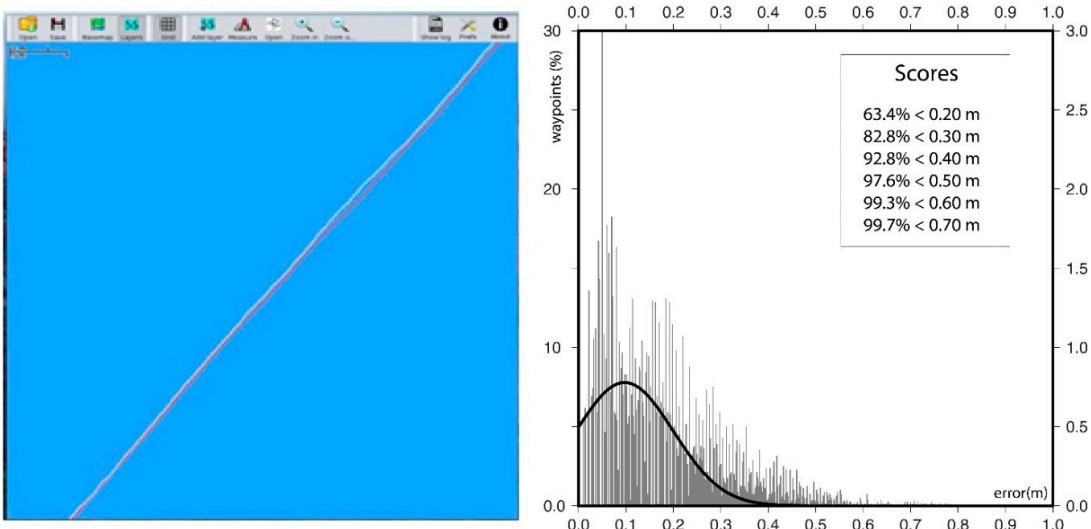

**Figure 8.** Navigation performance test along a 1 km straight line. **Left**: screenshot of OSN with first (white) and second (red) navigation lines. **Right**: distribution of the error modules between the two paths.

### 4.2. SBE

During the ASV developments, we considered the as SBE a "basic" sensor, since knowing water depth is crucial for safely navigating in shallow waters. Thus, we integrated a custom version of such instrument in the main electronic board of OpenSWAP. Our echosounder (µEcho) integrates a vertical incidence ultrasonic pinger, operating at a frequency of 200 kHz, a narrow (8°, conical) beam width, a short time-gate pulse length (350 µsec). The ultrasonic signal, is digitally sampled in a selected time-window by an ArduinoDue® board, and data (the echograms) are stored in SEGY-format files. Moreover, we implemented the procedure by Haynes et al., [14] to track the bottom reflection. First,

an amplitude envelope of the echogram is computed by convolving the squared values of the original data trace with a rectangular window as wide as the source pulse:

$$a(t) = \int_{-\infty}^{\infty} [x(\tau)]^2 w(t - \tau)\,d\tau \tag{1}$$

where $a(t)$ is the amplitude envelope trace, $x(t)$ is the recorded signal, is the rectangular window, and $L$ is the window size.

$$w(t) = \begin{cases} 1 & t \in [0, L] \\ 0 & elsewhere \end{cases}$$

Subsequently, a simple threshold-time delay algorithm was used to achieve the bottom reflected signal from the amplitude envelope trace. Conversion of travel-times into water-depth could be performed once the sound-speed is determined through specific estimates.

Acquisition of the entire echosounder sweep at each sounding point, rather than the simple depth value generally provided by echosounders, give us the opportunity of estimating the relative reflectivity of the sediment-water interface. Using the SeisPrho function RCCM [3], under the vertical incidence case and neglecting the effect of energy scattering due to the bottom roughness, we obtain an estimate of the relative reflection coefficient (R) by using:

$$R = (A_r/A_s)\,z \tag{2}$$

where $A_r$ and $A_s$ are the amplitudes of the source and reflected signals, respectively, and $z$ is the water depth.

In order to obtain an estimate of $R$ from our finite-length echo-sounder pulse, we used, for $A_s$ and $A_r$, the values:

$$A_s = \sum_{i=0}^{W} |\,x_i| \tag{3}$$

$$A_r = \sum_{i=B}^{B+W} |\,x_i| \tag{4}$$

where $x(i)$ is the digital sampled signal, $W$ is the width of the source pulse, and $B$ is the bottom detection time. Bottom reflectivity data obtained by these echograms are used to compile reflectivity maps, that could be diagnostic of geological processes.

Propagation and scattering of high frequency acoustic sound at or near the bottom is controlled by a number of factors, including biological, geological, biogeochemical and hydrodynamic processes operating at the benthic boundary layer [15]. However, experimental measurements suggest that the single most important geotechnical property related to acoustic attenuation is the mean grain size of the insonified sediment [16–19]. In such a case, a combination of bathymetric and reflectivity maps could be used as an effective tool in analyzing geological processes acting at the sediment-water interface. Figure 9 reports an example of such procedure carried out along a river stream. We note a main erosional trough formed close to a bridge pillar, also marked by high reflectivity (red pattern in Figure 9d). We also note that areas with prevailing deposition are marked by lower reflectivity (blue pattern in Figure 9d) as along the internal part of the river meander in the northern sector of the study area. Ground-truthing reflection coefficient estimates with sediment samples is mandatory to perform more accurate bottom classifications ([20] and references therein).

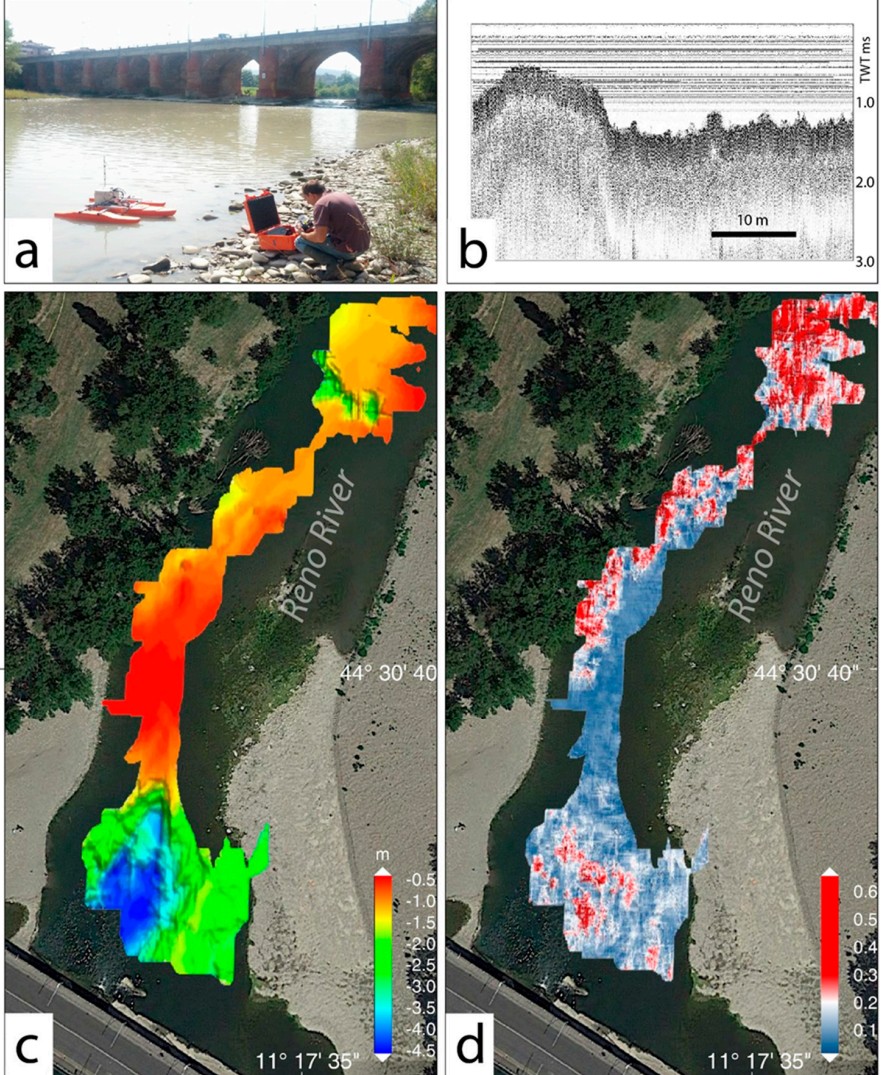

**Figure 9.** Echographic survey carried out along the Reno River close to an urban bridge (Bologna, Italy). (**a**) Configuration of the survey before start; (**b**) example of 200 kHz echogram in very shallow water; (**c**) morphobathymetric map highlighting the presence of a deep erosion close to the bridge pillar; (**d**) reflectivity map obtained using the same data showing sectors of prevalent erosion (red) and deposition (blue).

### 4.3. Side-Scan Sonar Imaging

We collected Side-Scan Sonar images along the Cavo Napoleonico artificial channel, in Northern Italy. The channel, which connects the Po and the Reno rivers in the Po plain, is oriented perpendicularly to the thrust and fold belt buried by alluvial sediments falling in the area which underwent the maximum superficial deformation during and after the Emilia 2012 earthquake sequence [21,22]. For this reason, it was chosen as an interesting site for geophysical surveys in search for co-seismic effects. Among other data collected with conventional methods, we carried out a side-scan sonar survey of this channel using a Starfish system mounted onboard of an OpenSWAP vehicle. We were searching for earthquake-related structures, such as fractures, fissures, sediment fluidization or slumps. Analysis of side-scan sonar images combined with high-resolution seismic reflection profiles (also collected with an OpenSWAP vehicle) suggests a correlation between the presence of "disturbances" at the channel floor (Figure 10) and the area o maximum deformation detected through satellite derived measures [23].

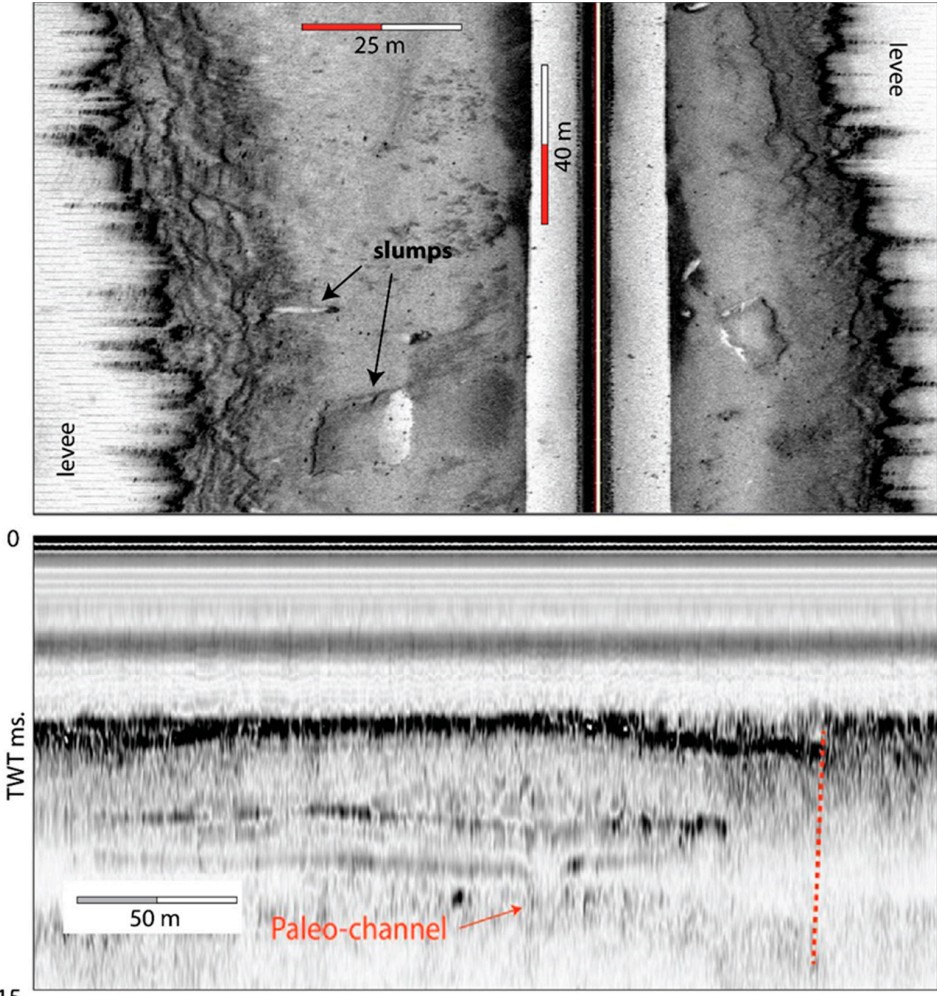

**Figure 10.** Top: side-scan sonar image of the Cavo Napoleonico artificial channel close to the epicenter of the Emilia 2012 earthquake, showing slumps and gravitative failures affecting the channel-floor. Bottom: SBP profile collected in the same area, penetrating the first meters of alluvial sediments, showing paleo-channels and displacements reaching up to the surface.

*4.4. High-Resolution Imaging of the Subsurface Using the Embedded SBP*

A typical SBP, operating with magnetostrictive transducers at hundreds to thousands of volts, is not suitable for operating on board of any OpenSWAP vehicle, either for the heavy weight and for low efficiency in converting the DC electric power of batteries to suitable high-voltages. For this reason, we developed a lightweight chirped SBP system (μCʜɪʀᴘ) embedded in the OpenSWAP electronics (see above). We tested the potential of our μCʜɪʀᴘ in Lake Trasimeno, a shallow-water tectonic lake in Central Italy. The lake was investigated through conventional systems in the frame of a geological study carried out with different geophysical methods [24,25]. However, some sectors close to the northern shore, were too shallow to be accessible using conventional systems, and were surveyed using OpenSWAP. The problem in this case was imaging at the best resolution the first tens of meters stratigraphic sequence in very shallow water, where several sources of noise affect in general the data. Prior to the survey, for a quality control of data collected by μCʜɪʀᴘ, we performed a comparison with an industry standard chirp-sonar system, the Benthos-Teledyne Chirp III, mounted onboard of a small boat. Results of this benchmark are reported in Figure 11, where the shallowest part (<10 ms. TWT) of the Trasimeno sedimentary sequence is imaged with the two systems, showing similar vertical resolutions and penetrations.

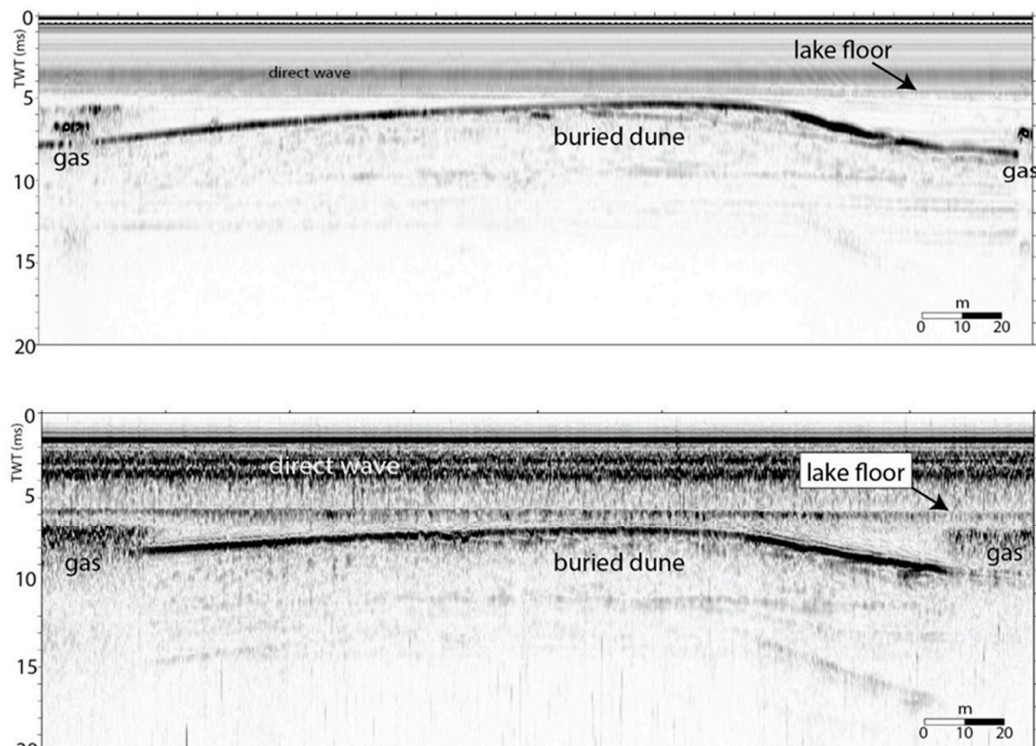

**Figure 11.** Example of seismic reflection profiles along the same navigation line collected with two different systems in Lake Trasimeno. **Top**: Chirp III Teledyne-Benthos, with 4 Massa transducers; **Bottom**: μChirp with 4 Monacor transducers. Unconsolidated sediments are penetrated down to 15–20 ms. below the lake floor, by both systems, with high vertical resolutions (tens of cm), enabling a detailed imaging the sedimentary structures.

### 4.5. Multibeam Echosounder Repeated Surveys

In order to test the possibility of performing MBES surveys using an OpenSWAP vehicle we used a Klein HydroChart 3500 integrated echosounder/side-scan sonar system. The HydroChart 3500 is a professional bathymetric sonar with IHO hydrographic standard for shallow water operations that integrates the characteristics of a side-scan sonar with those of an interferometric multibeam. It is a portable system that includes a motion reference unit (MRU) as well as course and sound speed sensors located in the sonar head. Each echogram includes uncertainty on the estimate of the depth and angles of the beams used for ray-tracing. In this way, the sonar propagation uncertainty model is integrated into the data processing flow, to provide uncertainty estimates for individual depth measurements that can be used by third-party bathymetric postprocessing.

We performed repeated multibeam survey offshore Calabria (Southern Italy), in the Calabrian Arc accretionary wedge, one of the most tectonically active regions in the Mediterranean Sea [26–28]. For this purpose, we surveyed repeatedly some key areas in the nearshore, where interferences between coastal sediment transport and gravitative instability in the vicinity of active faults were observed. The surveys were carried out in the same areas at different time-intervals, ranging from a few days to several months, obtaining good results. An example is reported in Figure 12, where we observe that over 95% of bathymetric measures are coherently positioned within ±30 cm of differences (the normal accuracy gathered by navigation system), and discrepancies between the two surveys (Figure 12c) are probably related to short-term seafloor changes, such as sand ripple migration, which were captured by our 4D survey. This first test indicates that repeating bathymetric measures at regularly spaced time-intervals along the same acquisition lines could be an interesting tool to determine what type of natural process is active and what is its temporal scale. Such information would be crucial in areas highly prone to slumping, seismogenic, and tsunamigenic risks.

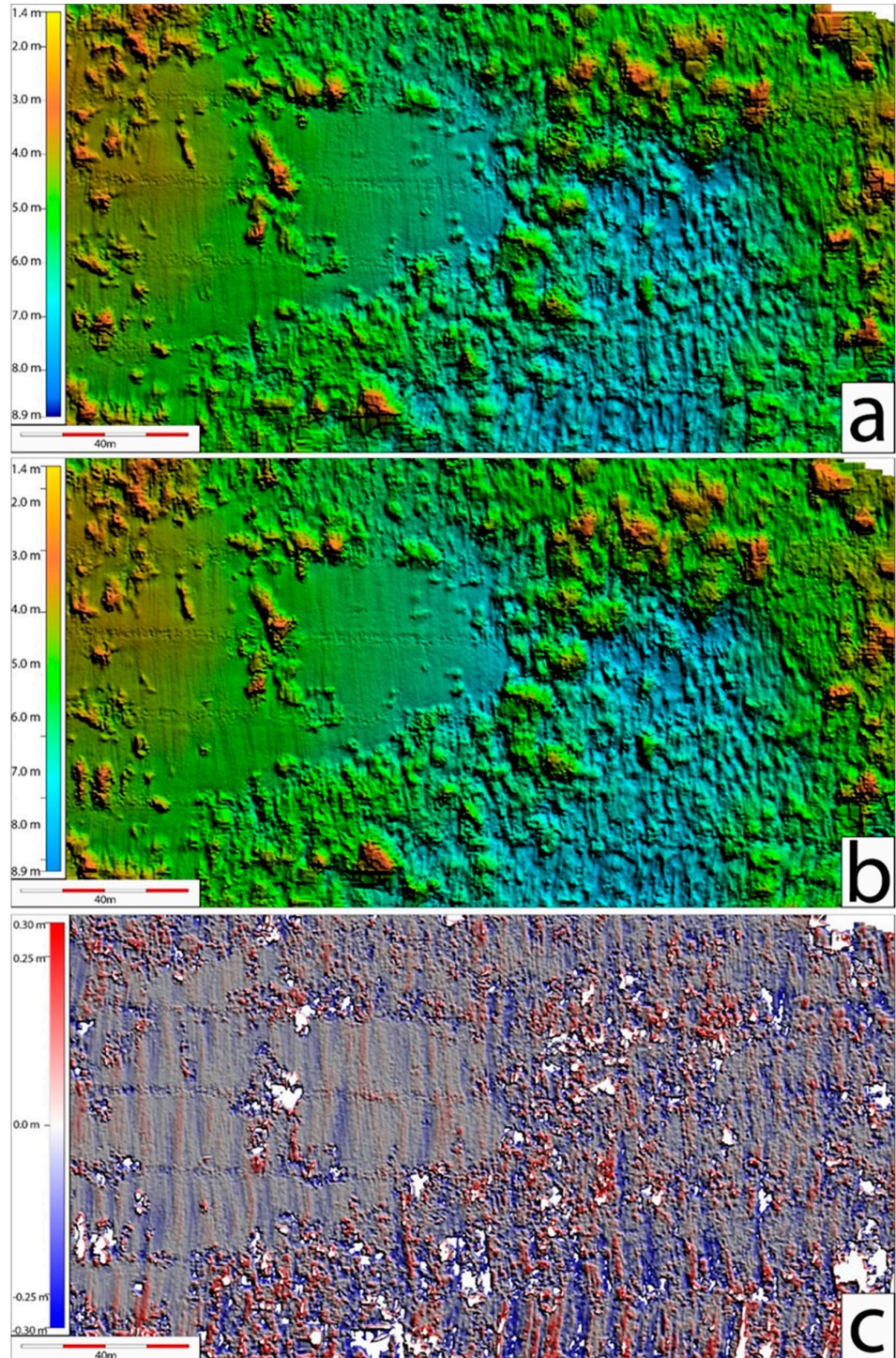

**Figure 12.** Bathymetric data collected using a Klein Hydrochart 3500 multibeam echosounder onboard of an OpenSWAP vehicle, including: (**a**) a first survey; (**b**) a second acquisition performed next day; and (**c**) the point-to-point difference between the two DTMs. Note that over 95% (colored dots) are within ±30 cm of difference. Red and blue undulations in c result from short-term seafloor changes, probably due to sand-ripple migration.

## 5. Discussion

The geophysical instruments to date installed onboard of the OpenSWAP vehicles are not exhaustive of all possible instruments that could be employed in geological, geophysical, geochemical,

and oceanographic studies generally carried out in shallow water areas. In fact, the possibility of installing instruments, such as electromagnetic sensors [29,30], or instruments for physical and chemical oceanography, could represent a dramatic improvement of their potential. The peculiar design of the hull is suitable for installing ADCPs (Acoustic Doppler Current-meters Profiler), that find a wide range of application in hydraulic and physical oceanographic studies. Furthermore, some basic chemical sensors, such as oxygen, nitrates, PH, etc., now available in small and light vessels, could represent interesting payloads to carry out rapid assessment of the surface water quality. We believe that the main characteristic of the OpenSWAP vehicles, i.e., low cost, high accuracy in performing predefined routes, embedded geophysical sensors, etc. could be already interesting for different scientific purposes. We have been able to verify that the high accuracy of the navigation algorithms would open the door to 4D surveys, i.e., surveys performed more than once in any given area, with the possibility of quantitatively comparing the results within acceptable error bars. Examples could be time-dependent analysis of morphological and stratigraphic changes of the sediment/water interface and shallow substratum eventually caused by the sediment dynamics (erosion vs. deposition), slumps and gravitative failures, earthquakes (slip along seismogenic faults and secondary effects of seismic shaking), tsunamis, etc.

Since the cost of the vehicles is maintained as low relative to other commercial systems, it would be possible to survey hazardous areas, accepting the high risks of damaging or losing the vehicles. Examples could be surveys carried out at glacier fronts, always prone to unpredictable collapses, or in environments characterized by strong currents and waves, such as rivers or coastlines, as well as in highly polluted environments, not safely accessed by manned vehicles or boats. The low cost would also enhance the use of small fleets of vehicles performing cooperative and adaptive surveys.

## 6. Conclusions

We tested the performances of OpenSWAP, an innovative class of autonomous surface vehicles (ASV) designed to carry out geophysical surveys in different shallow water environments. Such vehicles are based on open software and hardware architectures, to comply for low cost, high performances and easy customization. The possibility of carrying out surveys in shallow-water areas quickly, and at a fraction of costs relative to conventional methods, could disclose the use of these techniques to a wide range of users, allowing for the execution of repeated (4D) and cooperative missions, giving a time-variant perspective to the study of natural process in rapidly evolving environments. We note that most of the presented case studies were not feasible using conventional methods, and probably many others are waiting for such technologies to be tested. We hope that the widest availability of these vehicles would serve to promote the study of the coastal areas, and more generally the variety of shallow water environments, complex, rapidly evolving, and so important for their economic and naturalistic value.

**Supplementary Materials:** The following are available online at http://www.mdpi.com/2072-4292/12/16/2575/s1, Tables S1–S8.

**Author Contributions:** Conceptualization, L.G. and G.S.; methodology, L.G., G.S., and F.D.B.; software, G.S.; validation, L.G. and G.S.; formal analysis, G.S. and L.G.; investigation, L.G., G.S., and F.D.B.; resources, L.G.; data curation, L.G., G.S., and F.D.B.; writing—original draft preparation, L.G. and G.S.; writing—review and editing, L.G.; visualization, L.G.; supervision, L.G.; project administration, L.G.; funding acquisition, L.G. All authors have read and agreed to the published version of the manuscript.

**Funding:** This research was funded by POR-FESR Emilia Romagna Project NAIADI (P.I. Luca Gasperini).

**Acknowledgments:** The authors are grateful to all people who helped in different ways developing OpenSWAP in the frame of NAIADI and other projects. The list is long, but should include Alexandro Palmieri, Antonella Poggi, Federica Pasini, Gianni Biasini, Alessandro Giordano, Alfredo Liverani, Eugenio Nisini, Nicolò Marinelli, Francesco Suriano, Piero Zucchini, Valentina Ferrante and Alina Polonia.

**Conflicts of Interest:** The authors declare no conflict of interest.

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
