# Peer review of "OpenSWAP, an Open Architecture, Low Cost Class of Autonomous Surface Vehicles for Geophysical Surveys in the Shallow Water Environment"

_remotesensing, doi:10.3390/rs12162575_

Round 1
Reviewer 1 Report
Dear Authors
the paper about your OpenSWAP and his project is very interesting. You created a class of innovative open-architecture, low-cost autonomous vehicles for several surveys. Your system could solve many problems in very shallow geophysical acquisition.
the paper is well written and follows a good logical thread. you show two different cases that complete your work well. I suggest you:
1) To insert a table with the technical characteristics of the instrument, to facilitate understanding.
2) To explain better the sentence “under acceptable weather and sea conditions” … with some parameters
3) To insert the reference “Haynes et al. 1997” ; line 4234) To change in Lines 507-508 “first 15-20 ms” in < 10 ms … the other reflections (15-20 ms) are multiples
Congratulating you for your good work, I hope to see it operating in various geological/geophysical acquisitions
Best regards
Author Response
All requests of the referee were addressed, including:
1) a table with the technical characteristics of the instrument (Table 1) was inserted
2) we explained better what we mean by “under acceptable weather and sea conditions” (Line 95):
During our tests, we have found that the navigation path is followed up to Beaufort 4, but data quality under such conditions is very poor. We conclude that Beaufort 3 would be the limit for single-beam acoustic surveys.
3) We inserted the reference “Haynes et al. 1997” ; line 4234) and changed in Lines 507-508 “first 15-20 ms” in < 10 ms
Reviewer 2 Report
Dear Authors,
The article presents a new shallow-water survey vehicle. An appropriate vehicles for shallow-water geological and geophysical studies has always been of high demand. This challenging environment dictates the necessity for innovative approaches and methods. The presented vehicle appears to be a valuable and affordable solution for the researchers. As the paper is not just describing the engineering side of the vehicle but also is presenting some examples of the real-life applications in geophysical studies it makes it valuable to Remote Sensing.
Congratulations on a very good job in describing and presenting your technology and providing case studies and references. However, the paper requires careful editing.
The paper structure is unconventional which is due to the nature of the subject presented. The structure that authors chose follows the logic of the vehicle description and suits well for the purpose. However, some traditional elements could improve the paper.
My advise would be to add to the introduction some additional references and discuss examples for the studies that were carried out using shallow-water ASVs. Although the authors are presenting an innovative technology, there are commercial shallow-water ASVs like Z-boat as well as low-cost vehicles developed by other groups that already are being used for similar studies. That in my opinion would give a proper background for this study and better show the demand for the equipment.
The Section 4 - Performances of the vehicles and data acquisition examples - could be expanded to give more details for environmental setting of examples presented such as water depth, port/dock availability etc. Also Subsection 4.4. Sub-Bottom Profiler - is inconsistent with regards to other subsections with the study examples. It highlights a newly developed SBP system but not the results of the research that is used as an example. I would suggest to move the SBP description to Section 3 and talk more about the research itself in Section 4.4. The μECHO/μCHIRP part of Section 3, although is under a Software title, also describes the hardware which makes it hard for readers to understand if μCHIRP is a sensor or software (that is better explained in section 4). Overall, I would suggest to reconsider the structure of the Section 3 and maybe add a paragraph dedicated to a payload description.
Figure 8 the screenshot of OSN should be made a publishing quality. Especially the scale bar that is currently absolutely not readable.
Minor comments.
Careful editing of the section titles and formatting is required. Thus, the paper has two sections 5, three subsections 3.1., the section 4 is named 4.4, subsections 3.2 and 3.3 should be 3.3. and 3.4. respectively.
Subsections "Autonomous Navigation System'' and ''FORTH scripting language'' - italic.
line 14, 69 etc. ''morpho-bathymetric'' -> bathymetric
line 21, 54 etc. ''low-cost'' -> in this case ''low cost'' there are a number of unnecessary ''-'' throughout the text
line 52 '' the POR-FESR NAIADI, New autonomous/automatic systems for the study and monitoring of
aquatic environments'' -> ''the project New autonomous/automatic systems for the study and monitoring of
aquatic environments (POR-FESR NAIADI)''
line 72 ''Another important functionality
considered important'' -> ''Another functionality considered important''
line 95 etc. "multi-beam echosounders" -> "multibeam echosounders''
line 95 ''sub-bottom-profilers'' -> ''sub-bottom profilers''
line 114 "receive'' -> ''received''
line 284 ''Figure 5'' -> "Figure 6''
line 339 ''to modify to change'' -> ''to modify''
line 390 ''Figures 7'' -> "Figure 7''
line 451 move (4) to line 450
line 473 ''shoeing'' -> ''showing''
line 478 ''thrust-and fold'' -> ''thrust and fold''
lines 154, 183, 209, 229, 231,241, 292, 304, 324, 508 have acronyms that are not described in words. Note that the use of SCP/SBP acronym is inconsistent throughout the text and in one case (line 292) has a typo.
line 423 ''Haynes et al., (1997)'' there is no such reference in the list of references
line 467 "Ground-thruting reflection coefficient estimates with sediment samples is mandatory to
perform more accurate bottom classifications [20]'' - ''Ground-truthing''. Also this reference alone in the given sentence might be considered as an inappropriate self-citation as ground truth sampling for bottom classification is a well established procedure used for various methods. Thus, additional references are required.
lines 674-685 reference templates
The Supplementary Material requires a minor revisions for typos and formatting.
Author Response
My advise would be to add to the introduction some additional references and discuss examples for the studies that were carried out using shallow-water ASVs. Although the authors are presenting an innovative technology, there are commercial shallow-water ASVs like Z-boat as well as low-cost vehicles developed by other groups that already are being used for similar studies. That in my opinion would give a proper background for this study and better show the demand for the equipment.
We addressed this concern by including a new Table (Table 1) which list all peculiar characteristics of OpenSWAP relative to ASV availble in the market. This was also requested by Referee 1.
The Section 4 - Performances of the vehicles and data acquisition examples - could be expanded to give more details for environmental setting of examples presented such as water depth, port/dock availability etc.
We tried to expand such part inserting sentences which better explain the scientific context
Also Subsection 4.4. Sub-Bottom Profiler - is inconsistent with regards to other subsections with the study examples. It highlights a newly developed SBP system but not the results of the research that is used as an example. I would suggest to move the SBP description to Section 3 and talk more about the research itself in Section 4.4. The μECHO/μCHIRP part of Section 3, although is under a Software title, also describes the hardware which makes it hard for readers to understand if μCHIRP is a sensor or software (that is better explained in section 4).
We have been made minor changes trying to fulfill the requirements of the referees, but would like to keep this part as simple, since it would be really tedious describing the hardware characteristics of μECHO/μCHIRP, which we consider beyond the scope of this work, and possibly the subject of a further paper
Figure 8 the screenshot of OSN should be made a publishing quality. Especially the scale bar that is currently absolutely not readable.
YES, agreed. We modified the Figure according to the referee's suggestions
Minor comments.
Careful editing of the section titles and formatting is required. Thus, the paper has two sections 5, three subsections 3.1., the section 4 is named 4.4, subsections 3.2 and 3.3 should be 3.3. and 3.4. respectively.
Yes, thank you! There were problems of numbering (sorry). Now is ok
Subsections "Autonomous Navigation System'' and ''FORTH scripting language'' - italic.
OK, fixed
line 14, 69 etc. ''morpho-bathymetric'' -> bathymetric
OK, fixed
line 21, 54 etc. ''low-cost'' -> in this case ''low cost'' there are a number of unnecessary ''-'' throughout the text
We eliminated all unecessary "-"
line 52 '' the POR-FESR NAIADI, New autonomous/automatic systems for the study and monitoring of
aquatic environments'' -> ''the project New autonomous/automatic systems for the study and monitoring of
aquatic environments (POR-FESR NAIADI)''
OK, changed
line 72 ''Another important functionality
considered important'' -> ''Another functionality considered important''
OK, corrected
line 95 etc. "multi-beam echosounders" -> "multibeam echosounders''
OK, corrected
line 95 ''sub-bottom-profilers'' -> ''sub-bottom profilers''
OK, corrected
line 114 "receive'' -> ''received''
OK, corrected
line 284 ''Figure 5'' -> "Figure 6''
OK, corrected
line 339 ''to modify to change'' -> ''to modify''
OK, corrected
line 390 ''Figures 7'' -> "Figure 7''
OK, corrected
line 451 move (4) to line 450
OK, corrected
line 473 ''shoeing'' -> ''showing''
OK, corrected
line 478 ''thrust-and fold'' -> ''thrust and fold''
OK, corrected
lines 154, 183, 209, 229, 231,241, 292, 304, 324, 508 have acronyms that are not described in words. Note that the use of SCP/SBP acronym is inconsistent throughout the text and in one case (line 292) has a typo.
OK, we described acronyms and fixed the typo
line 423 ''Haynes et al., (1997)'' there is no such reference in the list of references
OK, added
line 467 "Ground-thruting reflection coefficient estimates with sediment samples is mandatory to
perform more accurate bottom classifications [20]'' - ''Ground-truthing''. Also this reference alone in the given sentence might be considered as an inappropriate self-citation as ground truth sampling for bottom classification is a well established procedure used for various methods. Thus, additional references are required.
OK, we referred to other works by adding [20 and references therein]. This part was also discussed before with appropiate referencing.
lines 674-685 reference templates
OK, deleted
The Supplementary Material requires a minor revisions for typos and formatting.
OK, done
Reviewer 3 Report
The mutual positioning of the hulls was optimal, noise-wise, at 3.7 knots. Authors did not specify what happens at different speeds.
Line 166: probably "described".
References to basic books, such as Stroustrup's "The C++ Programming Language" are definitely excessive.
Author Response
1) We added a sentence to describe what happens for speed higher that 3.7 knots, which is however the optimal speed for the surveys.
2) Line 166: probably "described".
OK corrected
2) We removed the pleonastic refernce to "Stroustrup's "The C++ Programming Language", that was replaced by a missing Haynes et al 1997, as pointed out by another referee.